# Senescence of endplate osteoclasts induces sensory innervation and spinal pain

**Dayu Pan, Kheiria Gamal Benkato, Xuequan Han, Jinjian Zheng, Vijay Kumar, Mei Wan, Junying Zheng, Xu Cao***

Department of Orthopedic Surgery and Department of Biomedical Engineering, Johns Hopkins University School of Medicine, Baltimore, United States

**Abstract** Spinal pain affects individuals of all ages and is the most common musculoskeletal problem globally. Its clinical management remains a challenge as the underlying mechanisms leading to it are still unclear. Here, we report that significantly increased numbers of senescent osteoclasts (SnOCs) are observed in mouse models of spinal hypersensitivity, like lumbar spine instability (LSI) or aging, compared to controls. The larger population of SnOCs is associated with induced sensory nerve innervation, as well as the growth of H-type vessels, in the porous endplate. We show that deletion of senescent cells by administration of the senolytic drug Navitoclax (ABT263) results in significantly less spinal hypersensitivity, spinal degeneration, porosity of the endplate, sensory nerve innervation, and H-type vessel growth in the endplate. We also show that there is significantly increased SnOC-mediated secretion of Netrin-1 and NGF, two well-established sensory nerve growth factors, compared to non-senescent OCs. These findings suggest that pharmacological elimination of SnOCs may be a potent therapy to treat spinal pain.

*For correspondence:
xcao11@jhmi.edu

## eLife assessment

This **fundamental** study advances our understanding of the role of senescent osteoclasts (SnOCs) in the pathogenesis of spine instability. The authors provide **compelling** evidence for the SnOCs to induce sensory nerve innervation. Accordingly, reduction of SnOCs by the senolytic drug Navitoclax markedly reduces spinal pain sensitivity. This work will be of broad interest to regenerative biologists working on spinal pain.

## Introduction

Low back pain (LBP) is the most common musculoskeletal problem globally, affecting at least 80% of all individuals at some point in their lifetime (*Chou et al., 2007*; *Deyo et al., 1991*; *Hoy et al., 2012*; *Maher et al., 2017*). It has also become the leading cause of years lived with disability worldwide, with almost 65 million cases involved per year (*Chen et al., 2022*; *James et al., 2018*), which consequently results in a tremendous medical burden and economic cost (*Deyo et al., 1991*; *Hartvigsen et al., 2018*). Current pharmacological treatment options for LBP include nonsteroidal anti-inflammatory drugs (NSAIDs), corticosteroids, opioids, etc. (*Qaseem et al., 2017*). However, the clinical use of these medications is limited due to potential severe adverse effects and modest therapeutic efficacy (*Roelofs et al., 2008*; *Deyo et al., 2015*; *Goldberg et al., 2015*). Some biological agents aiming for LBP management are under investigation as well (*Sheyn et al., 2019*; *Liao et al., 2019*). For example, parathyroid hormone has shown a superior antinociceptive effect on LBP in recent studies (*Chen et al., 2021*; *Nevitt et al., 2006*), whereas its risk of causing osteosarcoma and Paget's disease is not

negligible (*Andrews et al., 2012*; *Harper et al., 2007*; *Martin et al., 1976*). Thus, there is an urgent unmet clinical need for effective nonsurgical therapeutic interventions for LBP.

Cellular senescence is a stable and terminal state of growth arrest in which cells are unable to proliferate despite optimal growth conditions and mitogenic stimuli (*Di Micco et al., 2021*). It can be induced by various triggers, including DNA damage, telomere dysfunction, and organelle stress, and it has been linked to host physiological processes and age-related diseases, such as atherosclerosis, type 2 diabetes, and glaucoma (*Childs et al., 2016*; *He and Sharpless, 2017*; *Krishnamurthy et al., 2006*; *López-Otín et al., 2013*). Hence, the clearance of senescent cells has been suggested as a promising therapeutic strategy in several areas of pathology. For instance, glomerulosclerosis and decline in renal function in aged mice are rescued by clearance of p16[INK4a]-expressing senescent tubular brush-border epithelial cells, while clearance of senescent cells reduces age-related cardiomyocyte hypertrophy and improves cardiac stress tolerance (*Baker et al., 2016*).

Likewise, it has been shown that cellular senescence is an essential factor in the promotion of age-related musculoskeletal diseases, such as osteoporosis (*Farr et al., 2017*), osteoarthritis (OA) (*Jeon et al., 2017*), and intervertebral disc (IVD) degeneration (*Cherif et al., 2020*; *Wang et al., 2016*). The effectiveness of senolytic drugs towards bone-related diseases via elimination of senescent cells is also well documented (*Farr et al., 2017*). In particular, for the treatment of IVD degeneration, which is strongly associated with LBP, it was demonstrated that the senescent cells of degenerative discs were removed, and the IVD structure was restored by treatment with ABT263, a potent senolytic agent, in an injury-induced IVD degeneration rat model (*Lim et al., 2022*). Furthermore, recent research shows a direct link between telomere shortening-induced cellular senescence and chronic pain hypersensitivity (*Muralidharan et al., 2022*). Previous studies conducted by our laboratory have elucidated the significant role of osteoclasts in initiating the porosity of endplates with sensory innervation into porous areas and triggering LBP (*Ni et al., 2019*; *Xue et al., 2021*). Attenuating sensory innervation by inhibiting osteoclast activity could reduce spinal pain sensitivity. Importantly, it has been observed that some osteoclasts exhibit characteristics of senescence during osteoclastogeneses, such as the expression of p16 and p21, appearing to obtain a heterogeneous senescent phenotype (*Kwak et al., 2005*; *Gorissen et al., 2018*; *Mizoguchi et al., 2009*). Based on all these findings, we speculated there might be a subgroup of osteoclasts that are senescent (which we refer to as SnOCs), and they might promote the induction of sensory nerve innervation in porous endplates and spinal hypersensitivity.

Here, we demonstrate the presence of SnOCs in the porous endplates in two different spinal hypersensitivity mouse models induced by aging and LSI, respectively. We then deleted SnOCs in these models with ABT263, which resulted in a decreased number of tartrate-resistant acid phosphatase positive (TRAP[+]) OCs in the endplates along with decreased endplate porosity, reduced sensory innervation, and attenuated spinal pain behaviors. Together, these findings suggest the potential of utilizing senolytic drugs for the treatment of LBP and its associated pathologies.

## Results

### A significantly increased number of SnOCs are associated with endplate degeneration and spinal hypersensitivity in the LSI and aged mouse models

In this study, we used two different LBP mouse models created by LSI and aging. LSI was induced in 3-month-old C57/BL6 mice by surgically resecting the L3–L5 spinous processes along with the supraspinous and interspinous ligaments (*Bian et al., 2016*; *Ariga et al., 2001*; *Miyamoto et al., 1991*). Aged (24-month-old) C57BL/6J male mice were purchased from Jackson Laboratory. To explore spinal hypersensitivity, pain-related behavioral assessments, such as the von Frey test, hot plate test, and active wheel test, were performed on sham-operated mice, LSI mice, and aged mice. In both models, there was significantly less active time, distance traveled, and maximum speeds compared to the sham control (*Figure 1a–c*). Additionally, LSI mice and aged mice displayed significantly less reduced heat response times (*Figure 1d*), as well as significantly increased frequencies of paw withdrawal (PWF), depending on the strength of the mechanical stimulation, (*Figure 1e and f*) compared to the sham mice. By three-dimensional microcomputed tomography (μCT) analysis, we found a significant increase in the porosity and separation of trabecular bone (Tb.Sp) within the endplates of both LSI and aged mice compared to their younger counterparts without LSI (*Figure 1g and k*).

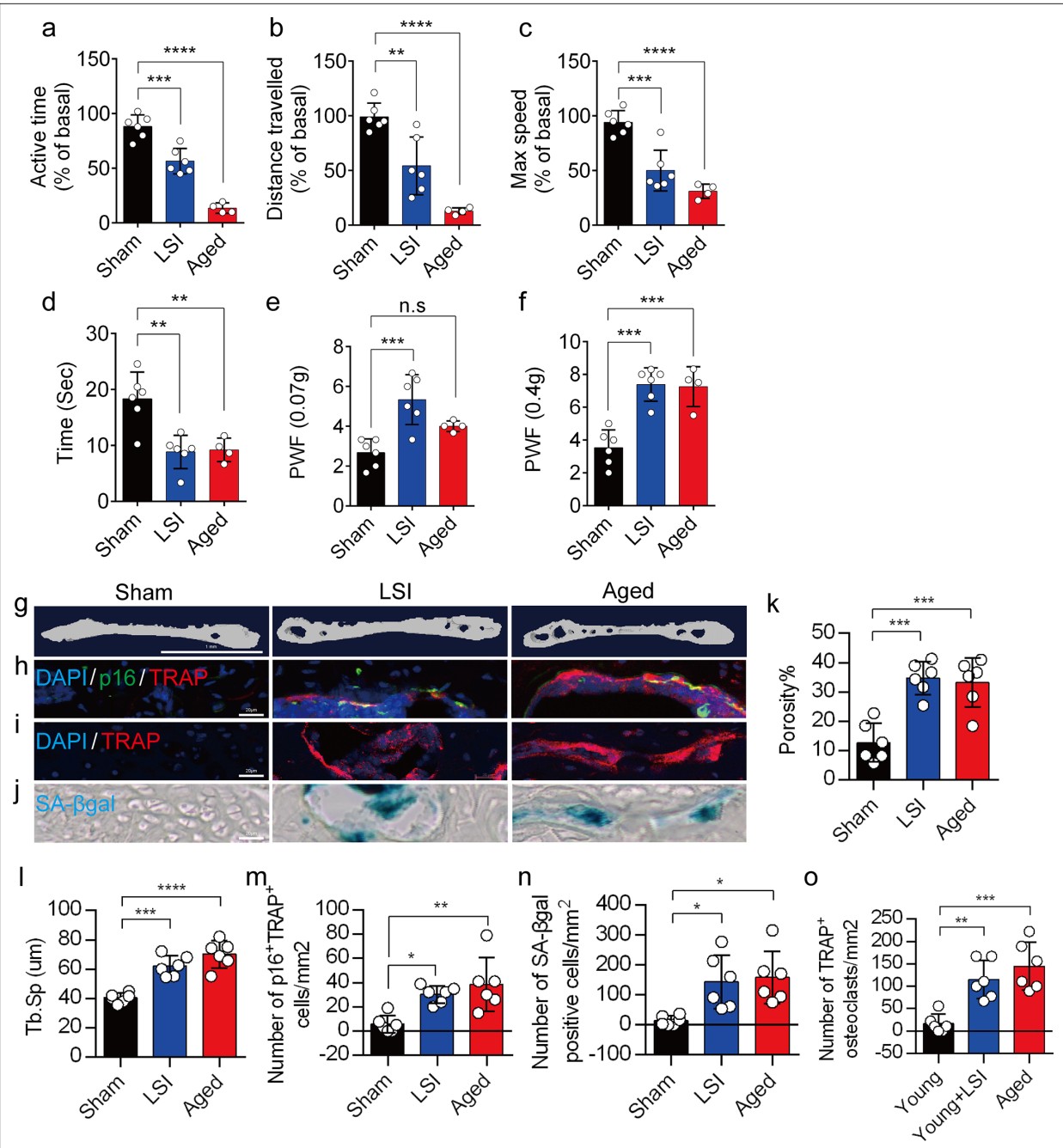

**Figure 1.** A greater number of senescent osteoclasts (SnOCs) are associated with endplate degeneration and spinal hypersensitivity in the lumbar spine instability (LSI) and aged mouse models. (**a–c**) Spontaneous activity, including active time (**a**), distance traveled (**b**), and maximum speed (**c**) on the wheel within 48 hr in the sham, LSI injury, and aged mice. (**d**) Time in seconds spent on a hot plate in the three groups of mice. (**e, f**) The frequency of hind paw withdrawal (PWF) in response to mechanical stimulation (von Frey test, 0.07 g (**e**) and 0.4 g (**f**)) in the sham, LSI injury, and aged mice. (**g**) Microcomputed tomography (μCT) images of coronal caudal endplate sections of L4–5 from 3-month-old sham and LSI and 24-month-old aged mice. (**h**) Immunofluorescent (IF) staining of p16 (green), tartrate-resistant acid phosphatase positive (TRAP) (red), and 4′,6-diamidino-2-phenylindole (DAPI) (blue) of the endplates of sham, LSI, and aged mice. (**i and j**) IF staining of TRAP (red) and DAPI (blue) (**i**) and senescence-associated beta-galactosidase (SA-βGal) (blue) staining (**j**) of endplate serial sections of sham, LSI surgery, and aged mice. (**k and l**) Microcomputed tomography (μCT) quantitative analysis of the porosity percentage (**k**) and trabecular separation (Tb.Sp) (**l**) of the endplates in the indicated groups. (**m**) Number of SnOCs (p16-positive and TRAP-positive cells) per mm2 in the indicated groups. (**n**) Number of SA-βGal (blue) positive cells per mm2 in the endplates in the indicated groups. (**o**) Number of TRAP (red) positive cells per mm2 in the endplates in the indicated groups. n≥4 per group. Scale bar, 1 mm (**g**) and 20 μm (**h, i, j**). Statistical significance was determined by one-way ANOVA, and all data are shown as means ± standard deviations.

The online version of this article includes the following figure supplement(s) for figure 1:

**Figure supplement 1.** Increased occurrence of SnOCs in the endplates of two LBP models compared to the control young sham mice.

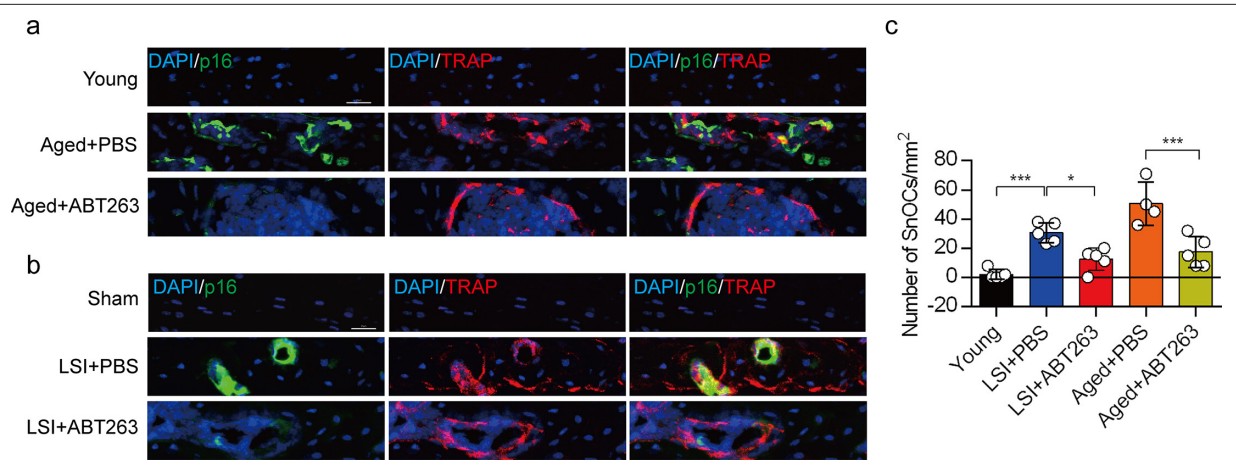

**Figure 2.** ABT263 effectively depletes endplate senescent osteoclasts (SnOCs) in the lumbar spine instability (LSI) and aging mouse models. (**a–c**) Immunofluorescent staining of p16 (green), tartrate-resistant acid phosphatase (TRAP) (red), and nuclei (4',6-diamidino-2-phenylindole [DAPI]; blue) of the endplates in aged (**a**) and LSI mice (**b**) injected with PBS (control) or ABT263 and the quantitative analysis of SnOCs based on dual staining for p16 and TRAP (**c**). n≥4 per group. Scale bar, 20 μm. Statistical significance was determined by one-way ANOVA, and all data are shown as means ± standard deviations.

To investigate the potential relationship between SnOCs and the degeneration of spinal endplates in the context of LSI and aging, co-staining of TRAP, a glycosylated monomeric metalloprotein enzyme expressed in osteoclasts, and p16, a tumor suppressor and established marker for cellular senescence, was performed (*Figure 1h*). We found that compared to the control young sham mice, there was a significantly increased number of p16+TRAP+ cells in the LSI and aged mice, indicative of SnOCs occurring in the endplates of these two mouse models (*Figure 1h and m*). To confirm the occurrence of SnOCs in the endplates in the LSI and aged mice, we stained the adjacent slides with TRAP and senescence-associated beta-galactosidase (SA-βGal) or HMGB1, markers for cellular senescence, respectively, and found that SnOCs existed in the two LBP mouse models but not in the sham controls (*Figure 1i, j, n, and o*, *Figure 1—figure supplement 1a and b*).

These findings collectively show a strong association between the presence of SnOCs and the development of spinal hypersensitivity, along with the degenerative changes in the endplates of mice subjected to LSI and during the aging process.

## ABT263 effectively depletes endplate SnOCs in the LSI and aging mouse models

To study the contribution of SnOCs to spinal pain, we first needed to show that such cells could be successfully depleted. Thus, we treated 24-month-old aged mice and 3-month-old sham and LSI mice with ABT263, a specific inhibitor targeting the anti-apoptotic proteins BCL-2 and BCL-xL, effectively leading to the depletion of SnOCs (*Chang et al., 2016*). ABT263 was administered via gavage at a dose of 50 mg/kg per day for 7 days per cycle, with two cycles separated by a 2-week interval, resulting in a total treatment period of 4 weeks. Remarkably, after the administration of ABT263, we observed a significant reduction of SnOCs in the endplates compared to the PBS-treated group (*Figure 2a–c*).

## Eliminating SnOCs reduces spinal hypersensitivity

To investigate spinal hypersensitivity, pain behavioral tests, including the von Frey test, hot plate test, and active wheel test, were conducted in sham, LSI, and aged mice treated with ABT263 and PBS, respectively. In the aged mice, ABT263 treatment resulted in a significant reduction in PWF (*Figure 3a and b*) and prolonged heat response times (*Figure 3c*) compared to the PBS-treated mice. In addition, there is a significant increased PWF in aged mice treated with PBS compared with young mice, particularly at 0.4 g instead of 0.07 g (*Figure 3a and b*). Furthermore, aged mice treated with PBS exhibited a significant reduction in both distance traveled and active time when compared to young mice (*Figure 3d and e*). Additionally, PBS-treated aged mice demonstrated a significantly shortened

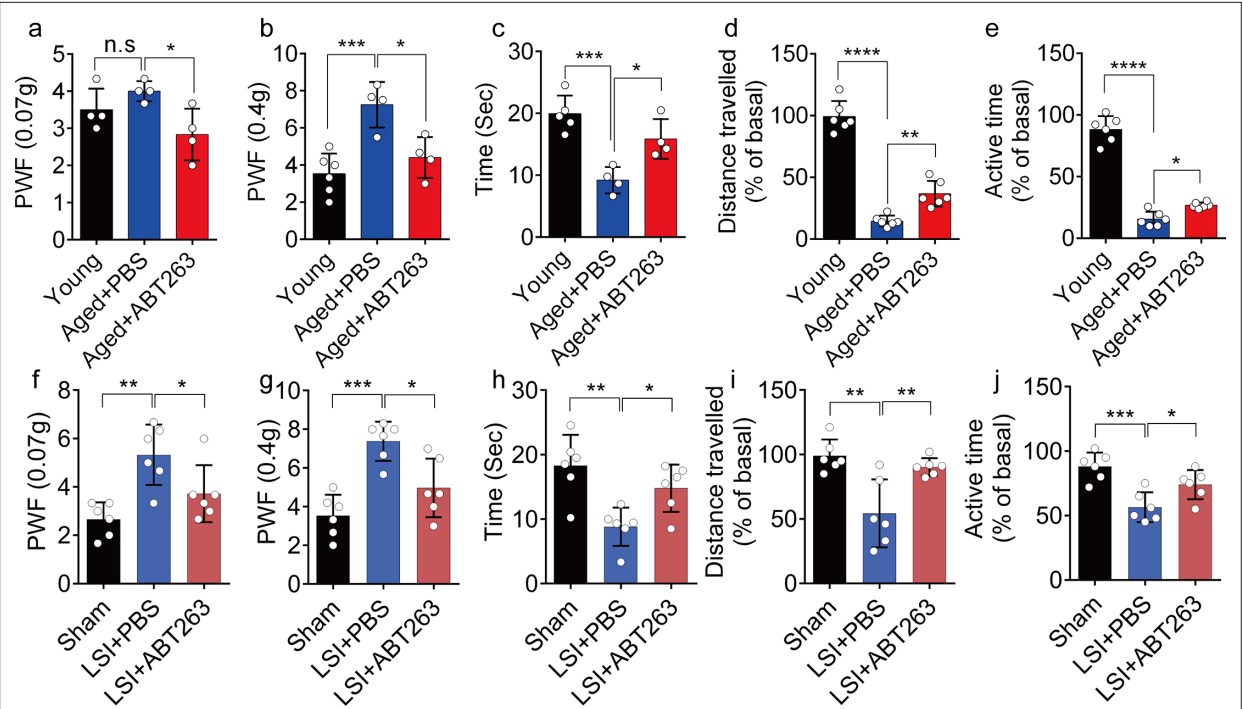

**Figure 3.** ABT263 treatment improves the symptomatic spinal pain behavior in the aged and lumbar spine instability (LSI) mouse models. (**a, b**) The paw withdrawal frequency (PWF) in response to mechanical stimulation (von Frey test, 0. 07 g (**a**) and 0.4 g (**b**)) in aged mice treated with PBS or ABT263 compared to young adult mice. (**c–e**) Time (in seconds) spent on a hot plate (**c**), as well as spontaneous activity, including distance traveled (**d**) and active time (**e**), on the wheel within 48 hr in aged mice treated with PBS or ABT263 compared to young adult mice. (**f, g**) The PWF in response to mechanical stimulation (von Frey test, 0. 07 g (**f**) and 0.4 g (**g**)) in the LSI mouse model treated with PBS or ABT263 compared to sham-operated mice. (**h–j**) Time (in seconds) spent on a hot plate (**h**), as well as spontaneous activity analysis, including distance traveled (**i**) and active time (**j**) on the wheel within 48 hr in the sham and LSI mice treated with PBS or ABT263. n≥4 per group. Statistical significance was determined by one-way ANOVA, and all data are shown as means ± standard deviations.

heat response time relative to young mice (*Figure 3c*). Importantly, these aged mice treated with ABT263 exhibited significantly increased distance traveled and active time compared to aged mice that received PBS injections (*Figure 3d and e*).

Notably, LSI mice treated with ABT263 also demonstrated substantial improvements across several parameters compared to the PBS-treated control mice. These improvements included lower PWF (*Figure 3f and g*), prolonged heat response time (*Figure 3h*), increased distance traveled (*Figure 3i*), and extended active time (*Figure 3j*). These results collectively indicate that the elimination of SnOCs reduces spinal hypersensitivity in both aged and LSI mouse models.

## Depletion of SnOCs reduces spinal degeneration and sustains endplate microarchitecture

To determine the effect of SnOCs on endplate architecture, degeneration, and osteoclast formation in the context of spinal pain in the aged mice, we conducted µCT analysis and immunostaining in aged mice treated with ABT263, or PBS and untreated 3-month-old young mice (*Figure 4a–e*). There was a significant reduction in endplate porosity and Tb.Sp of the caudal endplates of L4/5 in the aged mice treated with ABT263 compared to the PBS-treated aged group (*Figure 4f and g*). The PBS-treated aged mice exhibited a significant increase in endplate porosity (*Figure 4f*) and Tb.Sp (*Figure 4g*) compared to young mice. To examine the effects of ABT263 on endplate degeneration, we performed Safranin O staining and immunofluorescent staining to target matrix metalloproteinase 13-containing (MMP13[+]) and type X collagen-containing (ColX[+]) components within the endplate (*Figure 4b–d*). In the aged model, the ABT263-treated group exhibited a significant reduction in endplate score (*Figure 4h*), as well as the distribution of MMP13 and ColX within the endplates, compared to aged mice treated with PBS (*Figure 4i and j*). PBS-treated aged mice showed a significant elevation in

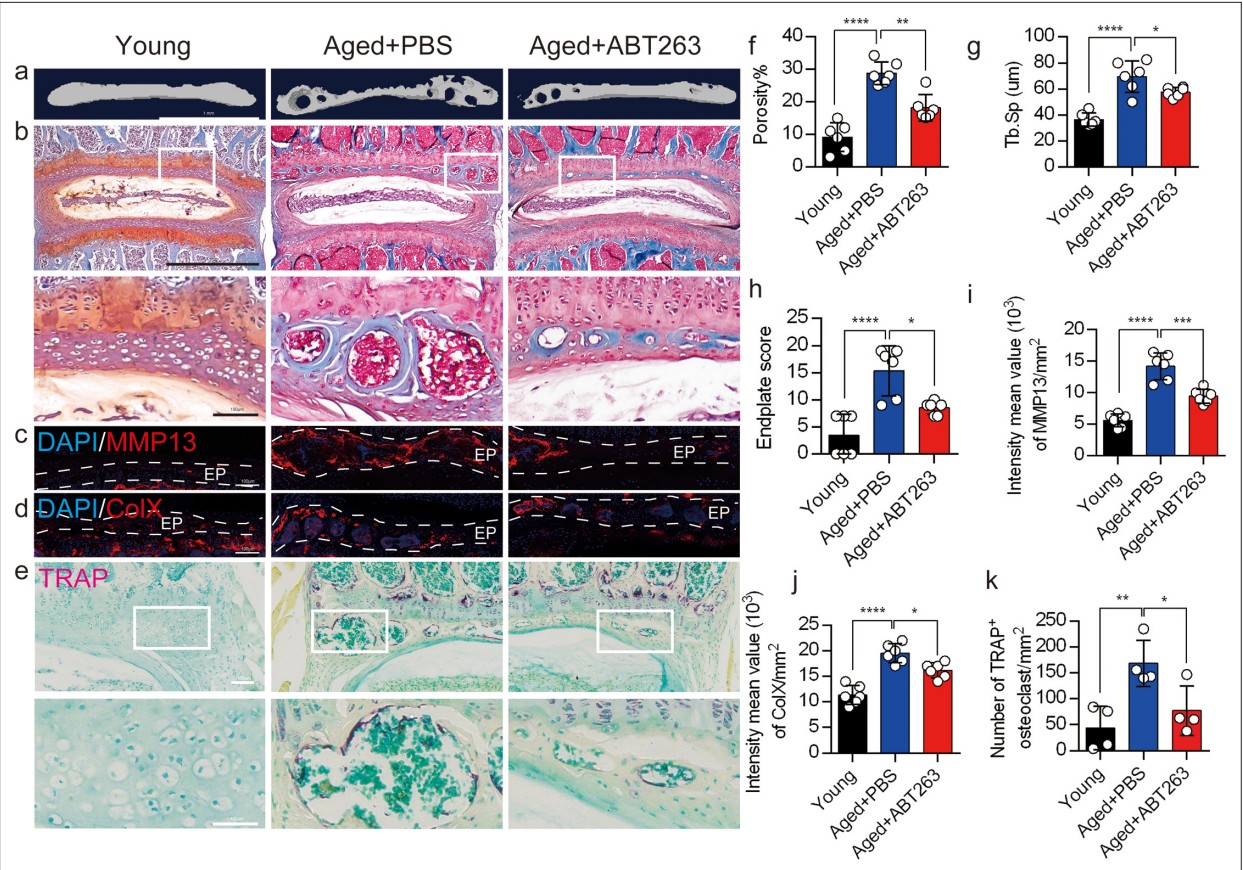

**Figure 4.** Depletion of senescent osteoclasts (SnOCs) reduces spinal degeneration and sustains endplate microarchitecture in aged mice.
(**a**) Microcomputed tomography (µCT) images of the aged mouse caudal endplates of L4–L5 injected with PBS or ABT263. Scale bar, 1 mm.
(**b**) Representative images of Safranin O and fast green staining of coronal sections of the caudal endplates of L4–5 in aged mice caudal endplates of L4–L5 injected with PBS or ABT263, respectively. Lower panners are zoomed in images from upper white boxes. Scale bar, 1 mm (upper panels) and 100 µm (lower panels). (**c**) Representative images of spine degeneration marker MMP13 (red) and nuclei (4',6-diamidino-2-phenylindole [DAPI]; blue) staining in aged mouse caudal endplates of L4–L5 injected with PBS or ABT263. Scale bar, 100 µm. (**d**) Representative images of spine degeneration marker ColX (red) and nuclei (DAPI; blue) staining in aged mouse caudal endplates of L4–L5 injected with PBS or ABT263. Scale bar, 100 µm.
(**e**) Representative images of tartrate-resistant acid phosphatase (TRAP) (magenta) staining of coronal sections of the caudal endplates of L4–5 in aged mice caudal endplates of L4–L5 injected with PBS or ABT263, respectively. Lower panners are zoomed-in images from upper white boxes. Scale bar, 100 µm. (**f**) The quantitative analysis of the porosity percentage. (**g**) The quantitative analysis of the trabecular separation. (**h**) The endplate score based on the Safranin O and fast green staining. (**i**) Quantitative analysis of the intensity mean value of MMP13 in endplates per mm2. (**j**) Quantitative analysis of the intensity mean value of ColX in endplates per mm2. (**k**) The quantitative analysis of the number of TRAP-positive cells in the endplate per mm2. n≥3 per group. Statistical significance was determined by one-way ANOVA, and all data are shown as means ± standard deviations.

endplate score (*Figure 4h*), as well as an increased distribution of MMP13 and ColX within the endplates when compared to young mice (*Figure 4i and j*). Furthermore, TRAP staining demonstrated a substantial rise in the count of TRAP+ osteoclasts within the endplates of aged mice in comparison to young control mice. Importantly, ABT263 treatment resulted in a significant reduction of TRAP+ osteoclasts within the endplates compared to PBS treatment (*Figure 4e and k*).

We next conducted µCT analysis and immunostaining in sham and LSI mice treated with ABT263 or PBS (*Figure 5a–e*) to determine the effect of SnOCs on endplate architecture, degeneration, and osteoclast formation in the context of spinal pain. In the LSI mice, ABT263 treatment significantly mitigated the porosity and Tb.Sp of the caudal endplates of L4/5 compared to the PBS-treated group (*Figure 5a, f, and g*). Safranin O staining and immunofluorescent staining of ColX and MMP13 within the endplate demonstrated ABT263 administration significantly reduced endplate score (*Figure 5b and h*), the distribution of MMP13 (*Figure 5c and i*) and ColX (*Figure 5d and j*) and TRAP+ osteoclasts (*Figure 5e and k*) in the endplates compared to LSI mice treated with PBS. These findings

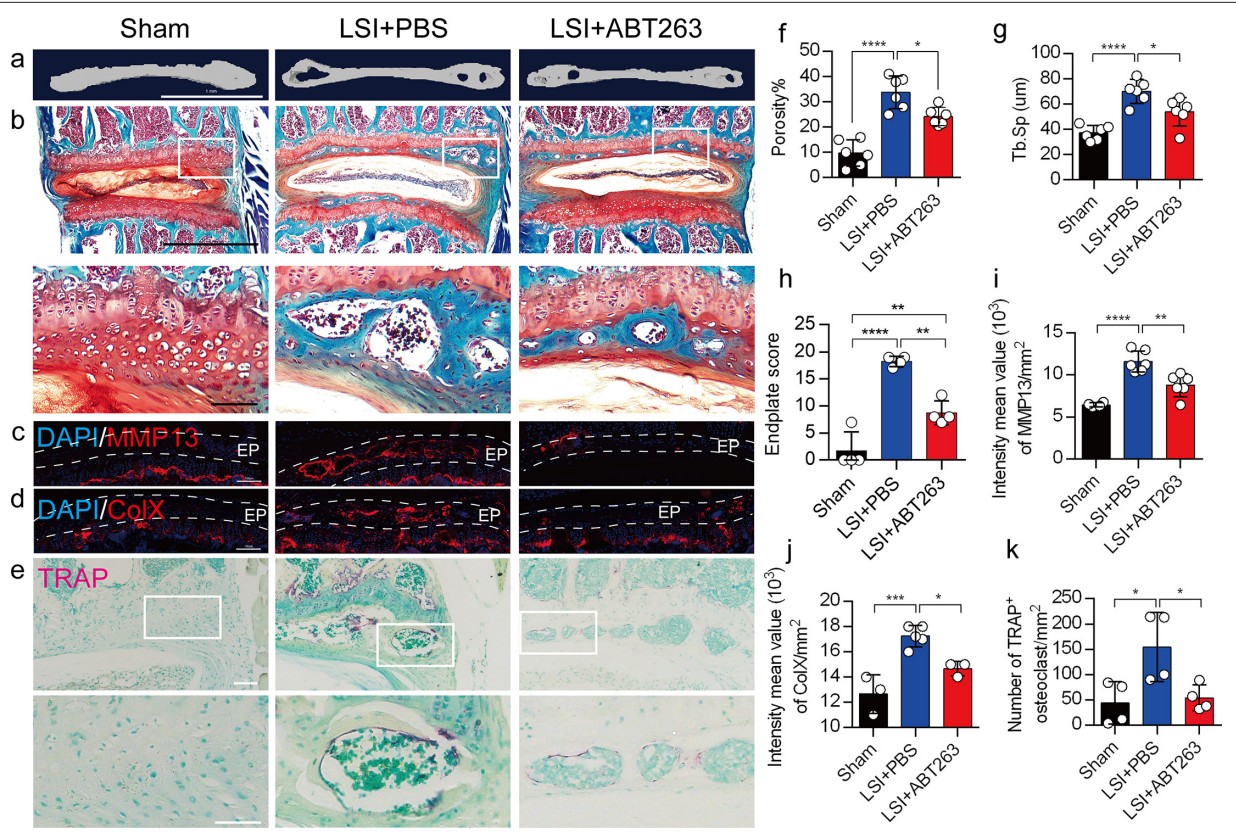

**Figure 5.** Depletion of senescent osteoclasts (SnOCs) reduces spinal degeneration and sustains endplate microarchitecture in lumbar spine instability (LSI) mice. (**a**) Microcomputed tomography (μCT) images of adult sham mice and 3-month-old LSI model mice caudal endplates of L4–L5 injected with PBS or ABT263. Scale bar, 1 mm. (**b**) Representative images of Safranin O and fast green staining in different groups. Lower panners are zoomed-in images from upper white boxes. Scale bar, 1 mm (upper panels) and 100 μm (lower panels). (**c**) Representative images of immunofluorescent staining of spine degeneration marker MMP13 (red) and nuclei (4′,6-diamidino-2-phenylindole [DAPI]; blue). Scale bar, 100 μm. (**d**) Representative images of immunofluorescent staining of spine degeneration marker ColX (red) and nuclei (DAPI; blue). Scale bar, 100 μm. (**e**) Representative images of tartrate-resistant acid phosphatase (TRAP) (magenta) staining in different groups. Lower panners are zoomed-in images from upper white boxes. Scale bar, 100 μm. (**f**) The quantitative analysis of the porosity percentage of the mouse caudal endplates of L4–5 measured by the μCT. (**g**) The quantitative analysis of the trabecular separation (Tb.Sp) of the mouse caudal endplates of L4–5 measured by the μCT. (**h**) The endplate score based on the Safranin O and fast green staining. (**i**) Quantitative analysis of the intensity mean value of MMP13 in endplates per mm2. (**j**) Quantitative analysis of the intensity mean value of ColX in endplates per mm2. (**k**) The quantitative analysis of the number of TRAP-positive cells in the endplate per mm2. n≥3 per group. Statistical significance was determined by one-way ANOVA, and all data are shown as means ± standard deviations.

collectively suggest the pivotal role of ABT263 in mitigating spinal degeneration and maintaining endplate remodeling in the context of spine pain.

## Depletion of SnOCs abrogates sensory innervation and pain

We previously found that sensory innervation occurs in the porous endplates, contributing to spinal hypersensitivity, in LSI and aged mice (*Ni et al., 2019*). To evaluate the contribution of SnOCs to sensory innervation and pain in LSI and aged mice, we co-stained for calcitonin gene-related peptide (CGRP), a marker of peptidergic nociceptive C nerve fibers, and PGP9.5, a broad marker of nerve fibers, in the endplates of LSI and aged mice treated with ABT263 or PBS (*Figure 6a*). Notably, we found fewer CGRP[+] PGP9.5[+] nerves in LSI mice and aged mice treated with ABT263 compared to those treated with PBS (*Figure 6b–e*).

To investigate the mechanisms underlying sensory innervation-induced spinal pain, we performed RT-qPCR to screen for expression of mediators regulating nerve fiber innervation and outgrowth, including Netrin-1 and NGF (*Bian et al., 2017*; *Papadakis et al., 2011*; *Rodriguez et al., 2012*). Compared to the young sham mice, aging and LSI was associated with significantly increased expression of *Ntn* and *Ngf* (the genes encoding netrin-1 and Ngf, respectively), which was substantially

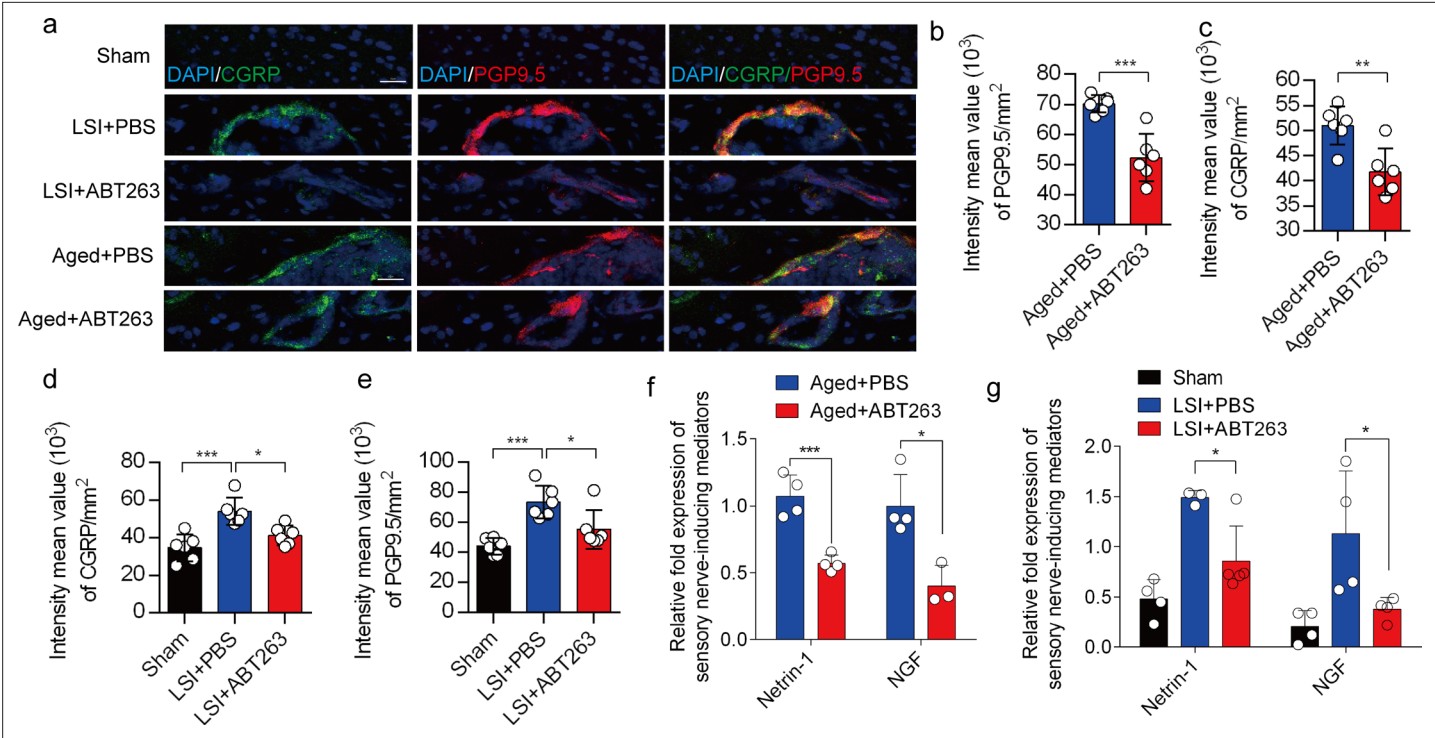

**Figure 6.** Depletion of senescent osteoclasts (SnOCs) abrogates sensory innervation and pain in aged and lumbar spine instability (LSI) mouse models. (a) Representative images of immunofluorescent analysis of calcitonin gene-related peptide (CGRP) (green), PGP9.5 (red), and nuclei (4′,6-diamidino-2-phenylindole [DAPI]; blue) of adult sham, LSI, and aged mice injected with PBS or ABT263. Scale bar, 20 μm. (b) Quantitative analysis of the intensity mean value of PGP9.5 per mm2 in aged mice. (c) Quantitative analysis of the intensity mean value of CGRP per mm2 in aged mice. (d) Quantitative analysis of the intensity mean value of PGP9.5 per mm2 in the LSI mouse model. (e) Quantitative analysis of the intensity mean value of CGRP per mm2 in the LSI mouse model. (f, g) Relative fold expression of *Ntn* and *Ngf* in aged mice (f) or LSI mice (g) with or without ABT263 treatment. n≥4 per group. Statistical significance in panels b, c, and f are analyzed using t-tests, while panels d, e, and g are subjected to one-way ANOVA. All data are shown as means ± standard deviations.

The online version of this article includes the following figure supplement(s) for figure 6:

**Figure supplement 1.** ELISA analysis of Netrin-1 and NGF in L3–5 endplates of sham, LSI+PBS, and LSI+ABT263 mice.

attenuated by ABT263 treatment in both LSI and aged mice (*Figure 6f and g*, *Figure 6—figure supplement 1a, b*).

Our earlier data showed that CGRP⁺ nociceptive nerve fibers and blood vessels were increased in the cavities of sclerotic endplates in the LSI and aged mice (*Ni et al., 2019*). To study whether elimination of SnOCs prevent such blood vessel growth into the endplates, we co-stained for CD31, an angiogenesis marker (green), and Emcn, an endothelial cells marker (red), in the endplates of sham, LSI, and aged mice treated with PBS or ABT263 (*Figure 7a*). In conjunction with the sensory nerve distribution within the porous endplates, we found noticeable growth of CD31⁺Emcn⁺ blood vessels into the endplates of the LSI and aged mice compared to young sham mice. This observation points toward an ongoing process of active ossification in the endplate. ABT263 treatment in the LSI and aged mouse models significantly mitigated the aberrant innervation of sensory nerves and blood vessels within the endplate compared to the PBS-treated mice (*Figure 7b–e*). Collectively, these findings underscore that ABT263 treatment effectively reduces spinal hypersensitivity by diminishing the innervation of sensory nerves and blood vessels within the endplate.

## Discussion

LBP affects individuals of all ages and is a leading contributor to disease burden worldwide. Despite advancements in its assessment and treatment, the management of LBP remains a challenge for researchers and clinicians alike. Defects in a number of anatomical structures within the back may be responsible for back pain, including the intervertebral discs, facet joints, muscles, ligaments, and

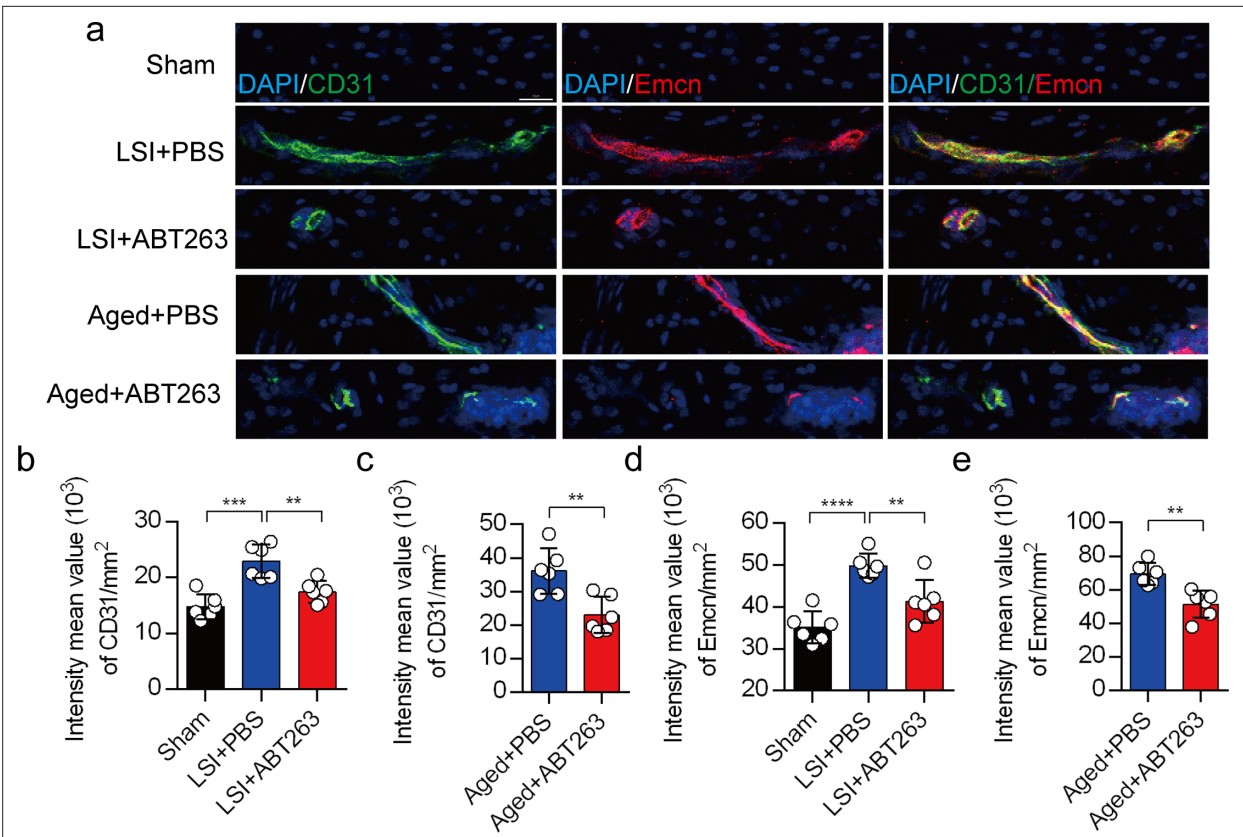

**Figure 7.** Depletion of senescent osteoclasts (SnOCs) abrogates blood vessels innervation in aged and lumbar spine instability (LSI) mouse models. (**a**) Representative images of immunofluorescent analysis of CD31, an angiogenesis marker (green), Emcn, an endothelial cell marker (red) and nuclei (4',6-diamidino-2-phenylindole [DAPI]; blue) of adult sham, LSI, and aged mice injected with PBS or ABT263. Scale bar, 20 μm. (**b**) Quantitative analysis of the intensity mean value of CD31 per mm2 in sham, LSI mice treated with PBS or ABT263. (**c**) Quantitative analysis of the intensity mean value of CD31 per mm2 in aged mice treated with PBS or ABT263. (**d**) Quantitative analysis of the intensity mean value of Emcn per mm2 in sham, LSI mice treated with PBS or ABT263. (**e**) Quantitative analysis of the intensity mean value of Emcn per mm2 in aged mice treated with PBS or ABT263. n≥4 per group. Statistical significance was determined by one-way ANOVA, and all data are shown as means ± standard deviations.

nerve root sheaths. Of these, the intervertebral discs, facet joints, and sacroiliac joints are implicated in the majority of the cases of LBP (*Kallewaard et al., 2010*). Furthermore, more than one structure may be contributing to the pain at any one time. During healing, neovascularization occurs and minute sensory nerves can penetrate the disrupted annulus and nucleus pulposus, leading to mechanical and chemical sensitization (*Zhen et al., 2022*).

During LSI or aging, endplates undergo ossification, leading to elevated osteoclast activity and increased porosity (*Bian et al., 2016*; *Bian et al., 2017*; *Papadakis et al., 2011*; *Rodriguez et al., 2012*). The progressive porous transformation of endplates, accompanied by a narrowed IVD space, is a hallmark of spinal degeneration (*Rodriguez et al., 2012*; *Taher et al., 2012*). Considering that pain arises from nociceptors, it is plausible that LBP may be attributed to sensory innervation within endplates. Additionally, porous endplates exhibit higher nerve density compared to normal endplates or degenerative nucleus pulposus (*Fields et al., 2014*). Netrin-1, a crucial axon guidance factor facilitating nerve protrusion, has been implicated in this process (*Hand and Kolodkin, 2017*; *Moore et al., 2012*; *Serafini et al., 1996*). The receptor mediating Netrin-1-induced neuronal sprouting, deleted in colorectal cancer (DCC), was found to co-localize with CGRP+ sensory nerve fibers in endplates after LSI surgery (*Forcet et al., 2002*; *Shu et al., 2000*). In summary, during LSI or aging, osteoclastic lineage cells secrete Netrin-1, inducing extrusion and innervation of CGRP+ sensory nerve fibers within the spaces created by osteoclast resorption. This Netrin-1/DCC-mediated pain signal is subsequently transmitted to the dorsal root ganglion (DRG) or higher brain levels. In our previous study, we found that osteoclasts induce sensory innervation of the porous areas of sclerotic endplates, which induced

spinal hypersensitivity in LSI-injured mice and in aging. Inhibition of osteoclast formation by knockout of *Rankl* in the osteocytes significantly inhibits LSI-induced porosity of endplates, sensory innervation, and spinal hypersensitivity (*Ni et al., 2019*). Likewise, knockout of *Ntn1* in osteoclasts abrogates sensory innervation into porous endplates and spinal hypersensitivity (*Ni et al., 2019*). In an OA mouse model, we found a role for osteoclast-secreted netrin-1 in the induction of sensory nerve axonal growth in the subchondral bone. Reduction of osteoclast formation by knockout of *Rankl* in osteocytes inhibited the growth of sensory nerves into subchondral bone, DRG neuron hyperexcitability, and behavioral measures of pain hypersensitivity (*Zhu et al., 2019*). Our previous study revealed that osteoclast-lineage cells may promote both nerve and vessel growth in osteoarthritic subchondral bone, leading to disease progression and pain (*Xie et al., 2014*). Here, we report that SnOCs are mainly responsible for modulating the secretion of netrin-1 and NGF, which mediate sensory innervation and induce hypersensitivity of spine.

Osteoclasts are the principal bone-resorbing cells essential for bone remodeling and skeletal development. Here, we report that osteoclasts in the endplate of the vertebral column undergo cellular senescence during injury and aging. The senescence process is programmed by a conserved mechanism because it is restricted to a specific region and follows a specific time course. Cellular senescence was defined by the presence of a senescence marker, SA-βGal, and a key senescence mediator, p16INK4a, detected in the bone tissue sections. In the present study, we found that the number of TRAP$^+$ and SA-βGal$^+$ or p16$^+$ senescent osteoclasts in endplates was significantly increased in LSI-injured mice and aged mice compared to sham-injured mice with PBS treatment. These findings support our hypothesis that increased numbers of SnOCs in LSI or aging conditions contribute to nerve innervation factors secretion, which leads to spine pain.

Current LBP management strategies have limited therapeutic effects, and progressive pathological spinal changes are observed frequently with these treatments. According to the American College of Physicians guidelines, pharmacological recommendations for acute or subacute LBP should begin with NSAIDs or muscle relaxants (moderate-quality evidence). There is no consensus on the duration of NSAID use, and caution is advised with persistent use due to concerns for cardiovascular and gastrointestinal adverse events. Guidelines by the American College of Physicians (*Qaseem et al., 2017*) recommend tramadol or duloxetine as a second-line treatment and opioids as the last-line treatment for chronic LBP. A meta-analysis showed that opioids offer only modest, short-term pain relief in patients with chronic LBP (*Abdel Shaheed et al., 2016*). The addictive potential of opioids coupled with several side effects limits their use in the management of such pain (*Qaseem et al., 2017*). Consequently, these drugs provide insufficient and unsustained pain relief with considerable adverse effects.

A clinical study demonstrated that nerve density is higher in porous endplates than in normal endplates and is associated with pain. Radiofrequency denervation treatment can be used for pain relief originating from the lumbar facet joints (*Lee et al., 2017*; *Maas et al., 2015*). However, National Institute for Health and Care Excellence (NICE) guidelines from the United Kingdom (*Bernstein et al., 2017*) recommend considering radiofrequency denervation only when the main source of pain originates from the facet joints, when pain is moderate to severe, and only when evidence-based multidisciplinary treatment has failed. During LBP progression, sensory nerves and blood vessels are aberrantly innervated in the endplate, which leads to pain. In this study, we aimed to reduce pain by decreasing nerve innervation. We found that SnOCs mediate nerve fiber innervation by elevated secretion of netrin-1 and NGF. Moreover, our previous study reported that osteoclasts can secret netrin-1 to attract sensory nerve growth (*Ni et al., 2019*). NGF can be produced by osteoblasts in response to mechanical load (*Kamei et al., 2022*) or by bone marrow stromal cells (BMSCs), and Sema3A is secreted by osteoblasts (*Tomlinson et al., 2017*). We believe a cross-talk exists between osteoclasts and osteoblasts or BMSCs after LSI or aging to modulate the secretion of nerve fiber innervation mediators.

The recent discovery that senescent cells play a causative role in aging and in many age-related diseases suggests that cellular senescence is a fundamental mechanism of aging. In aging, as well as in metabolic disorders, the immune response is affected by senescent cells that no longer replicate, but have a senescence-associated secretory phenotype that produces high levels of proinflammatory molecules. Certain chemotherapy drugs, known as senolytics, however, kill senescent cells and other drugs, called anti-SASP drugs, block their proinflammatory cell signaling. ABT263 is a selective BCL-2

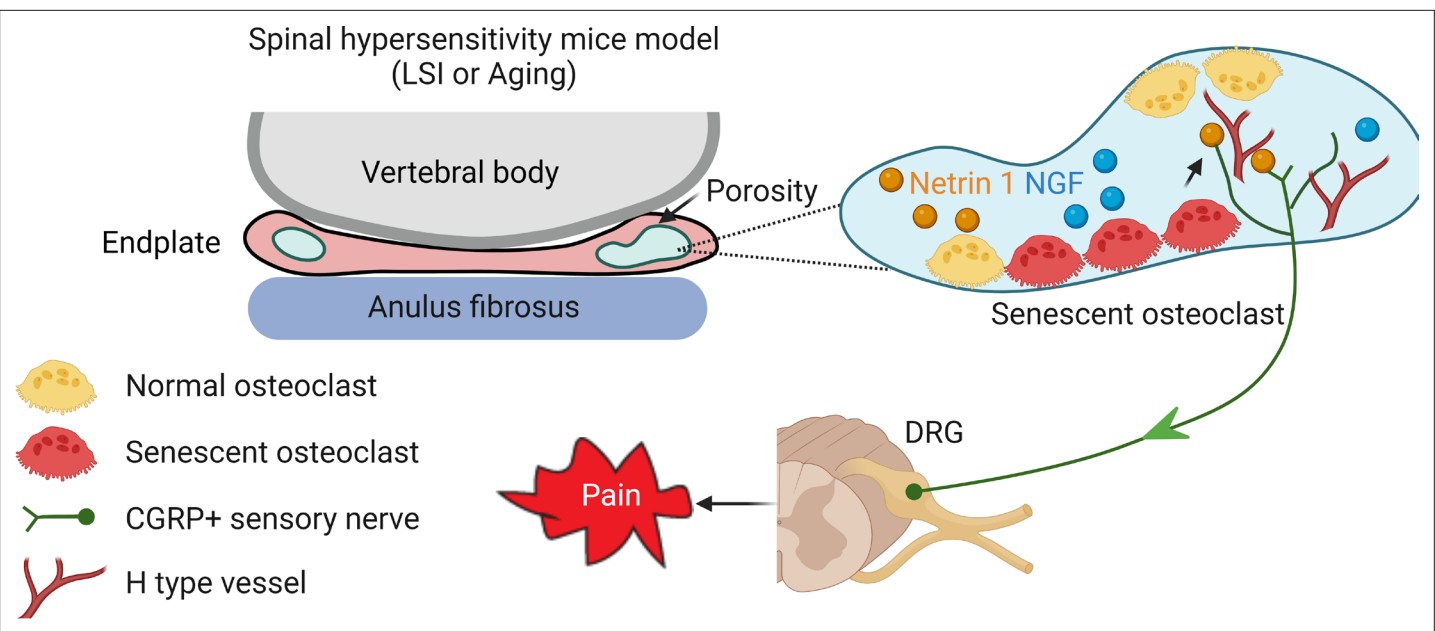

**Figure 8.** Schematic diagram of senescent osteoclasts (SnOCs) in porous endplate-induced spinal pain. In lumbar spine instability (LSI) or aging mouse models there is an induction of spinal hypersensitivity due to increased numbers of SnOCs in the endplate, leading to excessive secretion of sensory nerve mediators, such as Netrin-1, to attract calcitonin gene-related peptide (CGRP+) sensory nerve innervation. Additionally, aberrant microarchitecture and spinal degeneration are associated with increased wiring of sensory nerve fibers and H-type vessels in the endplate, all further contributing to lower back pain.

and BCL-xL inhibitor and is one of the most potent and broad-spectrum senolytic drugs. In the current study, we explored the role of ABT263 to manage spinal pain, and we found to it could effectively clear SnOCs cells in two mouse models. A decreased number of SnOCs will relieve pain by decreasing sensory neuron innervation in the endplate. However, ABT263 does not specifically eliminate SnOCs and thus further studies are required to prove the role of SnOCs in spinal hypersensitivity. Furthermore, ABT263 usually possesses various on-target and/or off-target toxicities, which could preclude its clinical use. Even so, for the first time to the best of our knowledge, we show evidence that SnOCs promote LBP by neurotrophic-mediated pathways. Our findings suggest that depletion of SnOCs, perhaps by use of a senolytic, can reduce sensory innervation and attenuate LBP, thus representing a new avenue in the management of this widespread condition (*Figure 8*).

## Methods
### Mice
Three- and 24-month-old C57BL/6J male mice were purchased from Jackson Laboratory. Three-month-old mice were anesthetized with ketamine (Ketalar, 0.13 mg/kg, intraperitoneally) and xylazine (Millipore Sigma; PHR3264, 12 mg/kg, intraperitoneally). Then, the L3–L5 spinous processes, and the supraspinous and interspinous ligaments were resected to induce instability of the lumbar spine and to create the LSI model (*Bian et al., 2016*; *Ariga et al., 2001*; *Miyamoto et al., 1991*). For mice in the sham group, we only surgically detached the posterior paravertebral muscles from L3 to L5. ABT263 (Navitoclax, Selleckchem, S1001) was administered to mice by gavage at 50 mg/kg per day for 7 days per cycle for two cycles with a 2-week interval between the cycles (the whole treatment time was 4 weeks). ABT263 was administered to aged (24 M) C57BL/6J mice (12 per group) at the age of 23 months. In the meantime, 4-month-old C57BL/6J mice 4 weeks post LSI or sham operation were treated with ABT263 or vehicle (PBS) (*Chang et al., 2016*). The control group (sham group) for the LSI group refers to C57BL/6J mice that did not undergo LSI surgery, while the control group (young group) for the aged group refers to 4-month-old C57BL/6J mice. All mice were maintained at the animal facility of The Johns Hopkins University School of Medicine (protocol number: MO21M276,

MO21M270, MO22M18). All experimental protocols were approved by the Animal Care and Use Committee of The Johns Hopkins University, Baltimore, MD, USA.

## Behavioral testing

Behavioral tests were performed after ABT263 administration and before sacrifice. All behavioral tests were performed by the same investigator, who was blinded to the study groups.

The hind PWF in response to a mechanical stimulus was determined using von Frey filaments of 0.07 g and 0.4 g (Aesthesio Precision Tactile Sensory Evaluator). Mice were placed on a wire metal mesh grid covered with a clear plastic cage. Acclimatization of the animals in the enclosure was for 30 min. We applied von Frey filaments to the mid-plantar surface of the hind paw through the mesh floor with enough pressure to buckle the filaments. A trial consisted of 10 times at 1 s intervals. Mechanical withdrawal frequency was calculated as the number of withdrawal times in response to 10 applications after three replicates.

The Hargreaves test of nociception threshold was evaluated by the Model Heated 400 Base. The animals were transferred from the holding room to the enclosure and acclimatization of the animals in the enclosure was for 30 min. The duration of exposure before the hind paw withdrawn with three replicates is recorded after focusing the mouse's mid-plantar surface of the hind paw with the light.

Spontaneous wheel-running activity was recorded using activity wheels designed for mice (model BIO-ACTIVW-M, Bioseb) (*Cobos et al., 2012*). The software enabled recording of activity in a cage similar to the mice's home cage, with the wheel spun in both directions. The device was connected to an analyzer that automatically recorded the spontaneous activity. We evaluated the distance traveled, maximum speed, and total active time during 2 days for each mouse.

## RT-qPCR

Mice were euthanized with an overdose of isoflurane inhalation. Total RNA was then extracted from the L4–L5 lumbar spine endplate tissue samples. Briefly, spine endplate tissue was ground in liquid nitrogen and isolated from Buffer RLT Plus using RNeasy Plus Mini Kits (QIAGEN, Germany) and homogenized directly with ultrasound (probe sonication at 50 Hz for three times, 10 s per cycle). The purity of RNA was measured by the absorbance at 260/280 nm. Then, the RNA was reverse transcribed into complementary DNA by PrimeScript RT (reverse transcriptase) using a primeScript RT-PCR kit (Takara), and qPCR was performed with SYBR Green-Master Mix (Thermo Fisher Scientific, USA) on a QuantStudio 3 (Applied Biosystems, USA). Relative expression was calculated for each gene by the 2-△△CT method, with glyceraldehyde 3-phosphate dehydrogenase (*GAPDH*) used for normalization. Primers used for RT-qPCR are listed below: *Netrin-1*: forward: 5'- CCTGTCACCTCT GCAACTCT -3', reverse: 5'- TGTGCGGGTTATTGAGGTCG -3'; *NGF*: forward: 5'- CTGGCCACACTG AGGTGCAT -3', reverse: 5'-TCCTGCAGGGACATTGCTCTC-3'; *BDNF*: forward:5'- TGCAGGGGC ATAGACAAAAGG -3', reverse: 5'- CTTATGAATCGCCAGCCAATTCTC -3'; *NT3*: forward: 5'- CTCA TTATCAAGTTGATCCA -3', reverse: 5'- CCTCCGTGGTGATGTTCTATT –3'; *Slit3*: forward: 5'- AGT TGTCTGCCTTCCGACAG -3', reverse: 5'- TTTCCATGGAGG GTCAGCAC -3'; *GAPDH*: forward: 5'-ATGTGTCCGTCGTGGATCTGA-3', reverse: 5'-ATGCCTGCTTCACCACCTTCTT-3'.

## ELISA

We determined the concentration of Netrin-1 (LSBio, LS-F5882) and NGF (Boster, EK0470) in the L3–L5 endplates using the ELISA Development Kit according to the manufacturer's instructions.

## µCT

Mice were euthanized with an overdose of isoflurane inhalation and flushed with PBS for 5 min followed by 10% buffered formalin perfusion for 5 min via the left ventricle. Then, the whole lumbar spine was dissected and fixed in 10% buffered formalin for 48 hr, transferred into PBS, and examined by high-resolution µCT (Skyscan1172). The scanner was set at a voltage of 55 kV, a current of 181 µA, and a resolution of 9.0 µm per pixel to measure the endplates and vertebrae. The ribs on the lower thoracic spine were included for identification of L4–L5 unit localization. Images were reconstructed and analyzed using NRecon v1.6 and CTAn v1.9 (Skyscan US, San Jose, CA, USA), respectively. Coronal images of the L4–L5 unit were used to perform three-dimensional histomorphometric analyses of the caudal endplate. The three-dimensional structural parameters analyzed were total porosity and

trabecular bone separation distribution (Tb.Sp) for the endplates. Six consecutive coronal-oriented images were used for showing three-dimensional reconstruction of the endplates and the vertebrae using three-dimensional model visualization software, CTVol v2.0 (Skyscan US).

## Histochemistry, immunohistochemistry, and histomorphometry

After µCT scanning, the spine samples were decalcified in 0.5 M EDTA (pH 7.4) for 30 days and embedded in paraffin or optimal cutting temperature compound (Sakura Finetek, Torrance, CA, USA).

Four-µm-thick coronal-oriented sections of the L4–L5 lumbar spine were processed for Safranin O (Sigma-Aldrich, S2255) and fast green (Sigma-Aldrich, F7252) staining, TRAP (Sigma-Aldrich, 387A-1KT) staining, and immunohistochemistry staining with an established protocol (*Ni et al., 2019*). Thirty-µm-thick coronal-oriented sections were prepared for blood vessel-related immunofluorescent staining, and 10-µm-thick coronal-oriented sections were used for other immunofluorescent staining.

The sections were incubated with primary antibodies to mouse Col X (1:100, ab260040, Abcam), MMP13 (1:100, ab219620, Abcam), Endomucin (1:100, sc-65495, Santa Cruz Biotechnology), CD31 (1:100, 550389, BD Biosciences), CGRP (1:100, ab81887, Abcam), PGP9.5 (1:100, SAB4503057, Sigma-Aldrich), Netrin-1 (1:100, ab39370, Abcam), TRAP (1:100, PA5-116970, Invitrogen), overnight at 4°C. Then, the corresponding secondary antibodies and 4',6-diamidino-2-phenylindole (Vector, H-1200) were added onto the sections for 1 hr while avoiding light.

The sample images were observed and captured by the confocal microscope (Zeiss LSM 780). ImageJ (NIH) software was used for quantitative analysis. We calculated endplate scores as described previously (*Boos et al., 2002*; *Masuda et al., 2005*).

## Statistics

All data analyses were performed using SPSS, version 15.0, software (IBM Corp.). Data are presented as means ± standard deviations. Unpaired, two-tailed Student's t-tests were used for comparisons between two groups. One-way ANOVA with Bonferroni's post hoc test was used for comparisons among multiple groups. For all experiments, $p < 0.05$ was considered to be significant. There were no samples or animals that were excluded from the analysis. The experiments were randomized, and the investigators were blinded to allocation during experiments and outcome assessment.

## Acknowledgements

This research was supported by the United States NIH National Institute on Aging under award numbers R01AG068997, P01AG066603, R01AG076783, R01AR071432 (to XC).

## Additional information

### Competing interests

Mei Wan: Reviewing editor, *eLife*. The other authors declare that no competing interests exist.

### Funding

| Funder | Grant reference number | Author |
| --- | --- | --- |
| National Institutes of Health | R01AG068997 | Xu Cao |
| National Institutes of Health | P01AG066603 | Xu Cao |
| National Institutes of Health | R01AG076783 | Xu Cao |
| National Institutes of Health | R01AR071432 | Xu Cao |

The funders had no role in study design, data collection and interpretation, or the decision to submit the work for publication.

## Author contributions
Dayu Pan, Conceptualization, Data curation, Software, Formal analysis, Validation, Investigation, Visualization, Methodology, Writing - original draft, Writing - review and editing; Kheiria Gamal Benkato, Formal analysis, Methodology; Xuequan Han, Data curation, Methodology; Jinjian Zheng, Validation; Vijay Kumar, Conceptualization; Mei Wan, Conceptualization, Supervision; Junying Zheng, Supervision, Writing - review and editing; Xu Cao, Conceptualization, Supervision, Funding acquisition, Project administration, Writing - review and editing

## Author ORCIDs
Dayu Pan (iD) http://orcid.org/0000-0003-0284-9350
Xuequan Han (iD) http://orcid.org/0000-0001-5629-5918
Mei Wan (iD) http://orcid.org/0000-0001-9404-540X
Junying Zheng (iD) http://orcid.org/0009-0001-2631-8055
Xu Cao (iD) http://orcid.org/0000-0001-8614-6059

## Ethics
All mice were maintained at the animal facility of The Johns Hopkins University School of Medicine (protocol number: MO21M276, MO21M270, MO22M18). All experimental protocols were approved by the Animal Care and Use Committee of The Johns Hopkins University, Baltimore, MD. All surgery was performed under ketamine and xylazine anesthesia, and every effort was made to minimize suffering.

Reviewer #1 (Public Review): https://doi.org/10.7554/eLife.92889.3.sa1
Reviewer #2 (Public Review): https://doi.org/10.7554/eLife.92889.3.sa2
Reviewer #3 (Public Review): https://doi.org/10.7554/eLife.92889.3.sa3
Author response https://doi.org/10.7554/eLife.92889.3.sa4

# Additional files

## Supplementary files
• MDAR checklist

## Data availability
All data generated or analyzed during this study are included in the manuscript and Mendeley Data (https://doi.org/10.17632/m4hhfk3tw7.1 and https://doi.org/10.17632/m4hhfk3tw7.2).

The following datasets were generated:

| Author(s) | Year | Dataset title | Dataset URL | Database and Identifier |
|---|---|---|---|---|
| Pan D, Benkato K, Han X, Zheng J, Kumar V, Wan M, Zheng J, Cao X | 2024 | Senescence of endplate osteoclasts induces sensory innervation and spinal pain | https://doi.org/10.17632/m4hhfk3tw7.1 | Mendeley Data, 10.17632/m4hhfk3tw7.1 |
| Pan D, Benkato K, Han X, Zheng J, Kumar V, Wan M, Zheng J, Cao X | 2024 | Senescence of endplate osteoclasts induces sensory innervation and spinal pain | https://doi.org/10.17632/m4hhfk3tw7.2 | Mendeley Data, 10.17632/m4hhfk3tw7.2 |

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
