## [Editor Report · eLife assessment]

This **fundamental** study advances our understanding of the role of senescent osteoclasts (SnOCs) in the pathogenesis of spine instability. The authors provide **compelling** evidence for the SnOCs to induce sensory nerve innervation. Accordingly, reduction of SnOCs by the senolytic drug Navitoclax markedly reduces spinal pain sensitivity. This work will be of broad interest to regenerative biologists working on spinal pain.

---

## [Referee Report · Reviewer #1 (Public Review)]

Summary:

In this study, Pan DY et al. discovered that the clearance of senescent osteoclasts can lead to a reduction in sensory nerve innervation. This reduction is achieved through the attenuation of Netrin-1 and NGF levels, as well as the regulation of H-type vessels, resulting in a decrease in pain-related behavior. The experiments are well-designed. The results are clearly presented, and the legends are also clear and informative. Their findings represent a potential treatment for spine pain utilizing senolytic drugs.

Strengths:

Rigorous data, well-designed experiments as well as significant innovation make this manuscript stand out.

Weaknesses:

All my concerns have been well addressed, no further comments.

---

## [Referee Report · Reviewer #2 (Public Review)]

Summary:

This manuscript examined the underlying mechanisms between senescent osteoclasts (SnOCs) and lumbar spine instability (LSI) or aging. They first showed that greater numbers of SnOCs are observed in mouse models of LSI or aging, and these SnOCs are associated with induced sensory nerve innervation, as well as the growth of H-type vessels, in the porous endplate. Then, the deletion of senescent cells by administration of the senolytic drug Navitoclax (ABT263) results in significantly less spinal hypersensitivity, spinal degeneration, porosity of the endplate, sensory nerve innervation, and H-type vessel growth in the endplate. Finally, they also found that there is greater SnOC-mediated secretion of Netrin-1 and NGF, two well-established sensory nerve growth factors, compared to non-senescent OCs. The study is well conducted and data strongly support the idea.

---

## [Referee Report · Reviewer #3 (Public Review)]

Summary:

This research article reports that a greater number of senescent osteoclasts (SnOCs), which produce Netrin-1 and NGF, are responsible for innervation in the LSI and aging animal models.

Strengths:

The research is based on previous findings in the authors' lab and the fact that the IVD structure was restored by treatment with ABT263. The logic is clear and clarifies the pathological role of SnOCs, suggesting the potential utilization of senolytic drugs for the treatment of LBP. Generally, the study is of good quality and the data is convincing.

Weaknesses:

All my concerns have been well addressed, no further comments.

---

## [Author Response]

The following is the authors’ response to the original reviews.

**Reviewer #1 (Public Review):**
Summary:In this study, Pan DY et al. discovered that the clearance of senescent osteoclasts can lead to a reduction in sensory nerve innervation. This reduction is achieved through the attenuation of Netrin-1 and NGF levels, as well as the regulation of H-type vessels, resulting in a decrease in pain-related behavior. The experiments are well-designed. The results are clearly presented, and the legends are also clear and informative. Their findings represent a potential treatment for spine pain utilizing senolytic drugs.Strengths:Rigorous data, well-designed experiments as well as significant innovation make this manuscript stand out.Weaknesses:Quantification of histology and detailed statistical analysis will further strengthen this manuscript.I have the following specific comments.(1) Since defining senescent cells solely based on one or two markers (SA-β-gal and p16) may not provide a robust characterization, it would be advisable to employ another wellestablished senescence marker, such as γ-H2AX or HMGB1, to corroborate the observed increase in senescent osteoclasts following LSI and aging.

We value the comments provided by the reviewer. In accordance with your suggestion, we have performed co-staining of HMGB1 with Trap in Supplementary Figure 1 to corroborate the observed augmentation of senescent osteoclasts following LSI and aging.

**Author response image 1. sa4fig1:** 

(2) The connection between heightened Netrin-1 secretion by senescent osteoclasts following LSI or aging and its relevance to pain warrants thorough discussion within the manuscript to provide a comprehensive understanding of the entire narrative.

We appreciate the reviewer's insightful comments. We have thoroughly addressed the entire narrative in the revised manuscript, as outlined below:

During lumbar spine instability (LSI) or aging, endplates undergo ossification, leading to elevated osteoclast activity and increased porosity1-4. The progressive porous transformation of endplates, accompanied by a narrowed intervertebral disc (IVD) space, is a hallmark of spinal degeneration4,5. Considering that pain arises from nociceptors, it is plausible that low back pain (LBP) may be attributed to sensory innervation within endplates. Additionally, porous endplates exhibit higher nerve density compared to normal endplates or degenerative nucleus pulposus6. Netrin-1, a crucial axon guidance factor facilitating nerve protrusion, has been implicated in this process7-9. The receptor mediating Netrin-1-induced neuronal sprouting, deleted in colorectal cancer (DCC), was found to co-localize with CGRP+ sensory nerve fibers in endplates after LSI surgery10,11. In summary, during LSI or aging, osteoclastic lineage cells secrete Netrin-1, inducing extrusion and innervation of CGRP+ sensory nerve fibers within the spaces created by osteoclast resorption. This Netrin-1/DCC-mediated pain signal is subsequently transmitted to the dorsal root ganglion (DRG) or higher brain levels.

(3) It appears that the quantitative data for TRAP staining in Figure 1j is missing.

We appreciate the reviewer's comments. We have added the statistical data of TRAP staining (Figure. 1p) to Figure 1 in the revised manuscript.

**Author response image 2. sa4fig2:** 

(4) Regarding Figure 6, could you please specify which panels were analyzed using a t-test and which ones were subjected to ANOVA? Alternatively, were all the panels in Figure 6 analyzed using ANOVA?

We appreciate the reviewer’s comments here. Upon careful review, we have ensured that quantitative data in panels b, c, and f are analyzed using t-tests, while panels d, e, and g are subjected to one-way ANOVA. These updates have been reflected in the revised figure legend.

**Reviewer #2 (Public Review):**
Summary:This manuscript examined the underlying mechanisms between senescent osteoclasts (SnOCs) and lumbar spine instability (LSI) or aging. They first showed that greater numbers of SnOCs are observed in mouse models of LSI or aging, and these SnOCs are associated with induced sensory nerve innervation, as well as the growth of H-type vessels, in the porous endplate. Then, the deletion of senescent cells by administration of the senolytic drug Navitoclax (ABT263) results in significantly less spinal hypersensitivity, spinal degeneration, porosity of the endplate, sensory nerve innervation, and H-type vessel growth in the endplate. Finally, they also found that there is greater SnOCmediated secretion of Netrin-1 and NGF, two well-established sensory nerve growth factors, compared to non-senescent OCs. The study is well conducted and data strongly support the idea. However, some minor issues need to be addressed.(1) In Figure 2C, "Number of SnCs/mm2", SnCs should be SnOCs.

We apologize for the oversight. This has been rectified in the revised manuscript.

**Author response image 3. sa4fig3:** 

(2) In Figure 3A-E, is there any statistical difference between groups Young and Aged+PBS?

We appreciate the reviewer's comments. Following your recommendation, we conducted additional statistical analyses to compare the young and PBS-treated aged mice, and we have incorporated these findings into the revised manuscript. The data reveals a significant increased paw withdrawal frequency (PWF) in aged mice treated with PBS compared with young mice, particularly at 0.4g instead of 0.07g (Figure 3a, 3b). Moreover, aged mice treated with PBS exhibited a significant reduction in both distance traveled and active time when compared to young mice (Figure. 3d, 3e). Additionally, PBS-treated aged mice demonstrated a significantly shortened heat response time relative to young mice (Figure. 3c).

**Author response image 4. sa4fig4:** 

(3) Again, is there any statistical difference between the Young and Aged+PBS groups in Figure 4F-K?

We appreciate the reviewer's comments. As per your suggestion, we conducted a thorough analysis to determine the statistical differences between the young and aged+PBS groups, and these statistical results have been implemented in the revised manuscript. The caudal endplates of L4/5 in PBS-treated aged mice exhibited a significant increase in endplate porosity (Figure. 4f) and trabecular separation (Tb.Sp) (Figure. 4g) compared to young mice.

Additionally, PBS-treated aged mice showed a significant elevation in endplate score (Figure. 4h), as well as an increased distribution of MMP13 and ColX within the endplates when compared to young mice (Figure. 4i, 4j). Furthermore, TRAP staining revealed a significant increase in TRAP+ osteoclasts within the endplates of PBS-treated aged mice as compared to young mice (Figure. 4k).

**Author response image 5. sa4fig5:** 

(4) What is the figure legend of Figure 7?

The legend for Figure 7 (as below) is included in a separate PDF file labeled 'Figures and Legends.' We have carefully checked the revised manuscript and made sure all the legends are included.

“Fig. 7. (a) Representative images of immunofluorescent analysis of CD31, an angiogenesis marker (green), Emcn, an endothelial cell marker (red) and nuclei (DAPI; blue) of adult sham, LSI and aged mice injected with PBS or ABT263. (b) Quantitative analysis of the intensity mean value of CD31 per mm2 in sham, LSI mice treated with PBS or ABT263. (c) Quantitative analysis of the intensity mean value of CD31 per mm2 in aged mice treated with PBS or ABT263. (d) Quantitative analysis of the intensity mean value of Emcn per mm2 in sham, LSI mice treated with PBS or ABT263. (e) Quantitative analysis of the intensity mean value of Emcn per mm2 in aged mice treated with PBS or ABT263. n ≥ 4 per group. Statistical significance was determined by one-way ANOVA, and all data are shown as means ± standard deviations. “

(5) In "Mice" section, an Ethical code is suggested to be added.

We appreciate the reviewer's comments. In accordance with your suggestion, we have included the Johns Hopkins University animal protocol number in the revised manuscript. The relevant paragraph has been updated to read: “All mice were maintained at the animal facility of The Johns Hopkins University School of Medicine (protocol number: MO21M276).”

(6) In "Methods" section, please indicate the primers of GAPDH.

We apologize for the absence of the GAPDH primers. Upon review, the GAPDH primers used were as follows: forward primer 5'-ATGTGTCCGTCGTGGATCTGA-3' and reverse primer 5'-ATGCCTGCTTCACCACCTTCTT-3'. These primer sequences have been included in the revised manuscript.

(7) Preosteoclasts are regarded to be closely related to H-type vessel growth, so do the authors have any comments on this? Any difference or correlation between SnCs and preosteoclasts?

The pre-osteoclast plays a crucial role in secreting anabolic growth factors that facilitate H-type vessel formation, osteoblast chemotaxis, proliferation, differentiation, and mineralization. The osteoclast represents the terminal differentiation phase, ultimately leading to the induction of resorption.

Senescent cells, including senescent osteoclasts, are characterized by permanent cell cycle arrest and changes in their secretory profile, which can impact their function. In the context of osteoclasts, senescence can lead to a reduction in bone resorption capacity and impaired bone remodeling. Senescent osteoclasts are believed to contribute to age-related bone loss and bonerelated diseases, such as osteoporosis.

**Reviewer #3 (Public Review):**
Summary:This research article reports that a greater number of senescent osteoclasts (SnOCs), which produce Netrin-1 and NGF, are responsible for innervation in the LSI and aging animal models.Strengths:The research is based on previous findings in the authors' lab and the fact that the IVD structure was restored by treatment with ABT263. The logic is clear and clarifies the pathological role of SnOCs, suggesting the potential utilization of senolytic drugs for the treatment of LBP. Generally, the study is of good quality and the data is convincing.Weaknesses:There are some points that can be improved:(1) Since this work primarily focuses on ABT263, it resembles a pharmacological study for this drug. It is preferable to provide references for the ABT263 concentration and explain how the administration was determined.

Thank you for your comment. ABT263 has been extensively employed in diverse research studies12-15. The concentration and administration of ABT263 followed the protocol outlined in the published paper13. The reference on how to use ABT263 is cited in the method section: “ABT263 was administered to mice by gavage at a dosage of 50 mg per kg body weight per day (mg/kg/d) for a total of 7 days per cycle, with two cycles conducted and a 2-week interval between them39”.

(2) It would strengthen the study to include at least 6 mice per group for each experiment and analysis, which would provide a more robust foundation.

Thank you for your comment here. In response, we conducted a new set of experiments, augmenting the majority of the sample size to six, and updated the corresponding statistical data in the revised manuscript.

(3) In Figure 4, either use "adult" or "young" consistently, but not both. Additionally, it's important to define "sham," "young," and "adult" explicitly in the methods section.

Thank you for your comment. We have addressed the inconsistency in the labeling of Figure 4. Additionally, we have explicitly defined "sham," "young," and "adult" in the methods section as follows: The control group (sham group) for the LSI group refers to C57BL/6J mice that did not undergo LSI surgery, while the control group (young group) for the Aged group refers to 4-month-old C57BL/6J mice.

**Author response image 6. sa4fig6:** 

(4) Assess the protein expression of Netrin 1 and NGF.

Thank you for your comment here. We employed ELISA to assess the protein expression of Netrin-1 and NGF in the L3 to L5 endplates. The data revealed that compared to the young sham mice, LSI was associated with significantly greater protein expression of Netrin1 and NGF, which was substantially attenuated by ABT263 treatment in LSI mice (Supplementary Fig. 2a, 2b)

**Author response image 7. sa4fig7:** 

Reference

(1) Bian, Q. et al. Excessive Activation of TGFbeta by Spinal Instability Causes Vertebral Endplate Sclerosis. Sci Rep 6, 27093, doi:10.1038/srep27093 (2016).

(2) Bian, Q. et al. Mechanosignaling activation of TGFbeta maintains intervertebral disc homeostasis. Bone Res 5, 17008, doi:10.1038/boneres.2017.8 (2017).

(3) Papadakis, M., Sapkas, G., Papadopoulos, E. C. & Katonis, P. Pathophysiology and biomechanics of the aging spine. Open Orthop J 5, 335-342, doi:10.2174/1874325001105010335 (2011).

(4) Rodriguez, A. G. et al. Morphology of the human vertebral endplate. J Orthop Res 30, 280-287, doi:10.1002/jor.21513 (2012).

(5) Taher, F. et al. Lumbar degenerative disc disease: current and future concepts of diagnosis and management. Adv Orthop 2012, 970752, doi:10.1155/2012/970752 (2012).

(6) Fields, A. J., Liebenberg, E. C. & Lotz, J. C. Innervation of pathologies in the lumbar vertebral end plate and intervertebral disc. Spine J 14, 513-521, doi:10.1016/j.spinee.2013.06.075 (2014).

(7) Hand, R. A. & Kolodkin, A. L. Netrin-Mediated Axon Guidance to the CNS Midline Revisited. Neuron 94, 691-693, doi:10.1016/j.neuron.2017.05.012 (2017).

(8) Moore, S. W., Zhang, X., Lynch, C. D. & Sheetz, M. P. Netrin-1 attracts axons through FAK-dependent mechanotransduction. J Neurosci 32, 11574-11585, doi:10.1523/JNEUROSCI.0999-12.2012 (2012).

(9) Serafini, T. et al. Netrin-1 is required for commissural axon guidance in the developing vertebrate nervous system. Cell 87, 1001-1014, doi:10.1016/s0092-8674(00)81795-x (1996).

(10) Forcet, C. et al. Netrin-1-mediated axon outgrowth requires deleted in colorectal cancer-dependent MAPK activation. Nature 417, 443-447, doi:10.1038/nature748 (2002).

(11) Shu, T., Valentino, K. M., Seaman, C., Cooper, H. M. & Richards, L. J. Expression of the netrin-1 receptor, deleted in colorectal cancer (DCC), is largely confined to projecting neurons in the developing forebrain. J Comp Neurol 416, 201-212, doi:10.1002/(sici)1096-9861(20000110)416:2<201::aid-cne6>3.0.co;2-z (2000).

(12) Born, E. et al. Eliminating Senescent Cells Can Promote Pulmonary Hypertension Development and Progression. Circulation 147, 650-666, doi:10.1161/CIRCULATIONAHA.122.058794 (2023).

(13) Chang, J. et al. Clearance of senescent cells by ABT263 rejuvenates aged hematopoietic stem cells in mice. Nat Med 22, 78-83, doi:10.1038/nm.4010 (2016).

(14) Lim, S. et al. Local Delivery of Senolytic Drug Inhibits Intervertebral Disc Degeneration and Restores Intervertebral Disc Structure. Adv Healthc Mater 11, e2101483, doi:10.1002/adhm.202101483 (2022).

(15) Yang, H. et al. Navitoclax (ABT263) reduces inflammation and promotes chondrogenic phenotype by clearing senescent osteoarthritic chondrocytes in osteoarthritis. Aging (Albany NY) 12, 12750-12770, doi:10.18632/aging.103177 (2020).